# The Effects of Postmortem Time on Muscle Trout Biochemical Composition and Structure

**DOI:** 10.3390/foods12101957

**Published:** 2023-05-11

**Authors:** Arno Germond, Annie Vénien, Christine Ravel, Brayan Castulovich, Jacques Rouel, Morgane Hutin, Sara Mezelli, Sandy Lefin, Pierre-Sylvain Mirade, Thierry Astruc

**Affiliations:** UR370, QuaPA, INRAE, 63122 Saint-Genès-Champanelle, France

**Keywords:** trout, processed fish, *postmortem*, spectroscopy, sustainable, machine learning

## Abstract

Fish industry operators have to process fish that arrive at various *postmortem* times. *Postmortem* time constrains processing and impacts product quality, safety, and economic value. The objective identification of biomarkers is desirable to predict the postmortem day of aging and this requires a comprehensive longitudinal characterisation of *postmortem* aging. We analysed the *postmortem* aging process in trout over a 15-day window. Quantitative physicochemical measurements (pH, colour, texture, a_w_, proteolysis, and myofibrillar protein solubility) performed on the same fish over time revealed the levels of protein denaturation, solubility, and pH, among other parameters, change very little when assessed by conventional chemical methods. Histological analyses were performed on thin sections and revealed fibre ruptures after 7 days of storage on ice. Ultrastructures were observed by transmission electronic microscopy (TEM) and revealed that sarcomere disorganisation occurred more often after 7 days of storage. Label-free FTIR micro-spectroscopy combined with a SVM model accurately predicted the *postmortem* time. Spectra-based PC-DA models also enable the identification of biomarkers corresponding to Day 7 and Day 15 *postmortem*. This study provides insights on *postmortem* aging and raises prospects for the rapid assessment of trout’s freshness status from label-free imaging.

## 1. Introduction

Trout is, behind salmon, the most consumed and farmed fish in Europe [1]. A substantial share of market trout is imported from the Atlantic and the North Sea, but a growing share is farmed and processed into a variety of products for the European market. Industry processors have several challenges to manage in their daily routine in order to guarantee high-quality products. One major issue is that the dates of fish arrivals vary, and suppliers often rely on qualitative methods to evaluate the *postmortem* time [2], and regular control for fillet spoilage [3]. Ideally, industry processors should adapt their processing (e.g., salting) according to *postmortem* time in order to maximise product quality and economical value or to minimise environmental impacts. Hence, the identification of biomarkers is desirable to predict the postmortem day of aging, and this requires a comprehensive longitudinal characterisation of *postmortem* aging while identifying the pertinent information and scale to perform the evaluation of the *postmortem* aging of fish.

Fish are prone to oxidation and the development of off-flavours due to processes that occur throughout *postmortem* aging. Aging is therefore considered a major factor in the freshness of fish fillets. In salmon, for example, *postmortem* aging affects yield, colour, texture, and gaping [4,5,6,7]. *Postmortem* time induces a water transfer in muscle fibres from the intracellular space to the extracellular space and then to outside the muscle, leading to a decrease in muscle fibre volume [8,9]. These changes are visible at the macroscopic scale and can therefore impact consumer choice and the downstream processing steps, such as salting. Concomitant with *postmortem* aging, proteolytic enzyme activity further contributes at varying degrees to an overall quality loss, in addition to microbially mediated processes. Biochemical analyses, such as the assessment of protein denaturation or lipid oxidation, and the measurements of macroscopic physical attributes, are conventionally performed in the industry.

Label-free imaging methods could enable industry processors to find biomarkers associated with fillet freshness or *postmortem* aging. Fast in situ characterisation of freshness, for example, using portable instruments, could help adjust the final salt concentration in filet. In fact, salting is often applied empirically by the industry with little consideration for the *postmortem* time, which can influence the salt diffusion in the muscles and impact the final salt concentrations. Pioneering studies first conducted the evaluation of freshness using the intrinsic fluorescence properties of proteins in salmon [10]. Later, fluorescence spectroscopy methods such as deep UV proved useful in estimating the aging process in animal products, such as meat, cheese, and milk [11]. Spectral imaging techniques combined with advanced analytical techniques for biomarker identification gained interest [12], yet the identification of biomarkers for the *postmortem* aging of fish is still limited. To the best of our knowledge, there is still no comprehensive study on the possibility of predicting accurately the postmortem time in trout.

To fill this gap, here we study several *postmortem* biochemical and structural characteristics of trout muscles during the 15-day period following animal slaughtering. In particular, we focus on several qualitative and quantitative parameters, such as pH, water activity, lipid oxidation, myofibrillar protein solubility, and proteolysis. We complemented our study of these parameters with histological observations, electron microscopy, and spectral imaging (FTIR) performed at the single fibre level. Using machine learning, we aimed to predict the postmortem time (seven time points) and also aimed to identify the biomarkers associated with *postmortem* aging and freshness in fish fillets. How these spectral biomarkers could be of use and relate to biochemical characteristics is also explored.

## 2. Materials and Methods

### 2.1. Sample Preparation

The trout fillets used for this study were prepared at an industrial site. Six farm-reared rainbow trout (*Oncorhynchus mykiss*) from the same batch were transported alive in a trailer to the processing industrial site located 90 km from the farm. The trout were stunned by electronarcosis and immediately bled, then stored for 15 min, eviscerated, weighed (2.87 ± 0.25 kg), placed in crushed ice, and rigorously identified. The trout were then transferred to a refrigerated room purposely made available to us to start the sampling procedure. To do so, a 1 cm-thick slice was cut near the head within 30 min of the time of death to serve for “Day 0” measurements and sampling. The fish were then put on ice, brought back to the laboratory, and kept on ice in a cold storage room (2 °C) until analysis. The fish temperature was monitored over 15 days and remained stable.

Previous research [10,13] states that salmon can be marketed until Day 15 after fishing, although industry processors will usually market trout up to Day 7. We performed a longitudinal study over 15 days. We took further 1 cm-thick slices at several time points post-kill by cutting the fish along the head-to-tail axis. Samples were taken at Day 0 *postmortem*, then on *postmortem* Days 1 (24 h), 3, 5, 7, 9, and 15 (see Figure 1). Each slice was divided into different areas, which we used for subsequent analyses. We assumed that the composition was homogenous within 1 slice. All solutions used below were provided by SIGMA (Sigma-Aldrich, St-Louis, MO, USA). Samples for FT-IR microspectroscopy were cryofixed in cooled isopentane and stored at −20 °C. Biochemical analyses were performed on six fish at each time point as described below. For electronic microscopy analyses and spectral analyses, measurements were performed on 3 fish randomly chosen, as described below. Samples for physicochemical analysis were immediately frozen and stored at −20 °C. For electron and light microscopy, 10 mm samples were chemically fixed by immersion in a solution of 2.5% glutaraldehyde in 0.1 M cacodylate buffer at pH 6.4 (close to the muscle pH) and stored at 4 °C until analysis.

### 2.2. Colour Measurements

Colour measurements were carried out on fresh samples for cutting, on six fish, at the three locations specified in Figure 1. The total sample number was 126 over the whole of the kinetic. Each sample was measured with 3 consecutive acquisitions, the values of which were averaged. Colours were measured based on the CIELab colour space system using a chroma meter (Konica Minolta CM-2500d, Tokyo, Japan) after calibration with a white standard. The trichromatic parameters L*, a*, and b* were calculated against CIE standard illuminant D65. The values were transferred directly to Excel using SpectraMagic NX software version 3.31 (Konica Minolta CM-2500d, Tokyo, Japan).

### 2.3. pH Measurements

The pH measurements were carried out on frozen samples at each aforementioned *postmortem* time point, on six fish, at three locations on each slice, over the kinetic (*n* = 126). To do so, 1 g of muscle was sampled and then ground in 10 mL of distilled water for 30 s at 15,000 rpm using a Polytron PT 2100 homogeniser (Kinematica, Luzern, Switzerland), and pH was measured three times in the suspension (Metrohm 654 pH-meter and Metrohm KCl 3 M pH electrode) (Metrohm, Herisau, Switzerland).

### 2.4. Water Activity (a_w_) Measurements

Water activity was measured in frozen samples at each aforementioned *postmortem* time point, for six fish, at three locations on each slice, over the kinetic (*n* = 126). Samples of around 3 to 5 g (cut into small pieces) were placed in a measuring cup and left for about 90 min to equilibrate. The a_w_ value was read using an a_w_ measurement device (AW-Sprint TH-500 Novasina, Garches, France) calibrated with standards certified to the following a_w_ values: 0.11, 0.33, 0.53, 0.75, 0.90, and 0.98.

### 2.5. Lipid Oxidation

Lipid oxidation was measured in frozen samples at each aforementioned *postmortem* time point, for six fish, and at a similar location on each slice. Lipid oxidation was evaluated using the T-BARS method, which measures malondialdehyde (MDA), a by-product of lipid peroxidation. First, 1 g of sample was ground in 10 mL of KCL 0.15M + BHT 0.1 mM for 30 s at 15,000 rpm (Polytron PT 2100), then 300 µL aliquots of ground samples were vortexed for 15 s with 150 µL TCA 2.8% and 150 µL TBA 1% in 50 mM NaOH before incubation in a boiling water bath for 10 min. After cooling at room temperature for 30 min, the pink chromogen was extracted by adding 1.2 mL 1-butanol, vortexing for 15 s, and centrifugation at 4000 rpm for 15 min at 4 °C. The absorbance of the organic phase was measured at 535 nm and 760 nm. T-BARS concentrations (mg of MDA per kg of sample) were calculated using 1,1,2,2-tetramethoxypropane as standard. Measurements were performed in triplicate.

### 2.6. Myofibrillar Protein Solubility

Soluble proteins were extracted from muscle samples following a modified protocol described by Santé-Lhoutellier et al. [14], which we adapted for 96-well plate measurements. Briefly, for each fish and at each time point, three 1-g samples were cut from the shown area (Figure 1), resulting in a total of 180 samples. Samples were homogenised in 10 mL of cooled phosphate buffer 0.025 mM (pH 7.2) using an Ultra-Turrax T10 IKA homogeniser at maximum speed for 30 s. The extracts were centrifuged at 1500 rpm for 20 min at room temperature and then left overnight in the dark at 4 °C. The next day, 200 µL supernatants were placed in test tubes and mixed with 300 µL CuSO_4_ 1%, 200 µL sodium cholate, and 3 mL NaOH at 10%. The tubes were closed, agitated for 30 min, then centrifuged at 4000 rpm for 15 min at 4 °C. Solubility was measured following a modified Biuret method [15]. Microplates were prepared to measure each sample in triplicate (*n* = 540), with 3 negative controls for each. For control samples, the sample solution was replaced with ultrapure water and 3 BSA samples at 10 mg/mL, which we used as standard. Optical density was determined at 540 nm and 760 nm, and the difference between the two measures was used to quantify the amount of myofibrillar protein. The amount of protein was calculated by comparing the obtained values to BSA measurements and expressed in mg/L.

### 2.7. Proteolytic Activity

Proteolytic activity was estimated by fluorescamine labelling of the N-terminal of amino groups of peptides and free amino acids, following a modified protocol described by [16], which we modified for 96-well plate measurements. The fluorescamine solution was prepared by dissolving 3.0 mg in 1.0 mL of DMSO in an amber tube and mixing by hand (30 s), and the solution was kept at ambient temperature and used within 24 h. For each time point, six slices of fish were analysed in triplicate, resulting in a total of 126 samples. For each cut sample, we weighed out 0.5 g of muscle and placed it in a polypropylene test tube containing 3 mL of sterilised Milli-Q water, and then homogenised the sample (T10 Ultra-Turrax IKA) at 18,000 rpm for 30 s at room temperature. Samples were then kept in the dark at 4 °C and used within an hour. In new tubes, 1600 µL of a 12.5% TCA solution was added with 400 µL of the ground sample or glycine solution. Borate buffer at a pH of 8.2 was used to give highly fluorescent derivatives, which we found better than using acetone. Samples were mixed for 15 min at 500 rpm at room temperature and centrifuged at 4500 rpm for 10 min at 4 °C. We then prepared 96-well plates. To measure the fluorescence value of each tube, we prepared 3 wells containing 100 µL of borate buffer, 100 µL of the sample, and 60 µL of fluorescamine solution in DMSO. We also prepared three negative control wells with borate, DMSO with no fluorescamine, and 100 µL of sample solution. This enabled us to later subtract the value of the negative controls from the value of the samples. Wells were mixed using an automated plate mixer, then incubated at room temperature for 20 min before measurements 5 min later. Fluorescence values were measured by fluorescence spectroscopy on a Jasco FP-8300 system (Jasco France, Lisses, France) at 472 nm using an excitation wavelength of 390 nm. The more proteins are hydrolysed, the more –NH_2_ is accessible in solution and can thus form a fluorescent complex in the presence of fluorescamine. We used glycine solutions prepared at 6 concentrations (final concentrations 0, 1.25, 2.5, 5, 10, and 25 mM) to establish a standard curve and estimate the amount of protein hydrolysed in the solution, expressed in mM equivalent to L-glycine. The glycine control was measured on each day of the experiment to ensure that our apparatus provided consistent readings.

### 2.8. Histological Observation by Light and Transmission Electron Microscopy

For light and electron microscopy, three fish (ID 1 to 3) were analysed over the whole kinetic to limit the time and costs of experiments. Small blocks (1 to 3 mm^3^) were cut from the strips previously immersed in the 2.5% glutaraldehyde fixative, and post-fixed in 1% osmium tetroxide in cacodylate buffer for 1 h at room temperature. The blocks were dehydrated through an increasing gradient of ethanol concentrations (70%, 95%, and 100%) then in acetone, and embedded in epoxy resin (TAAB, Eurobio France). Semi-thin muscle sections of 1 µm thickness cut longitudinally to fibre direction (Reichert-Jung ultramicrotome, Germany) were put on a glass slide and stained with toluidine blue. Images were acquired using an Olympus BX 61 microscope (Olympus, Tokyo, Japan) coupled with a high-resolution digital camera (Olympus DP 71) and Olympus Cell Sens software version 2.3 (Olympus France SAS, Rungis, France). From the epoxy resin-embedded samples initially used for semi-thin section preparation, ultrathin sections of 90 nm thickness cut longitudinally to fibre direction (Reichert-Jung ultramicrotome, Germany) were deposited on copper grids and stained with uranyl acetate and lead citrate for transmission electron microscopy (Reynolds, 1963). Samples were prepared at the INRAE research institute and observed at the Cellular Imaging Centre for Health (CICS) laboratory (Clermont-Ferrand University, France). Ultrathin sections were observed under a transmission electron microscope (Hitachi HM 7650, Tokyo, Japan) at 80 kV acceleration voltage. Micrographs were generated using a Hamamatsu AMT digital camera system (Hamamatsu, Japan) coupled with the microscope. Images were acquired at magnifications of ×3000, ×8000, ×12,000, and ×25,000. Sarcomere lengths were quantified using Image J software v.1.53r for Windows 10. At least 50 sarcomeres per fish were measured for D0, D5, D7, D9, and D15, taking care to make the measurements in different muscle fibres. For each sample, a mosaic was taken with the ×4 lens; then, images were taken with the ×10 and ×20 lenses to show any alterations in the samples in more detail.

### 2.9. Fourier Transform Infrared (FTIR) Measurements and Pre-Processing

The three fish selected for electron microscopy were also chosen for FTIR analyses. Muscle samples cryofixed in cooled isopentane (−160 °C) were cut in transverse orientation at −25 °C using a Leica CM1950 cryomicrotome (Nussloch, Germany). Tissue cryosections (6 µm depth) were placed onto BaF_2_ windows and analysed with a FTIR microscope (Thermo Scientific iN10, Thermo Fisher Scientific, Madison, WI, USA) equipped with a liquid nitrogen-cooled detector. After the acquisition of a mosaic image of the tissue, we targeted the centre of a single-fibre cell to measure it with a 30 µm aperture. At least 35 cells were measured per fibre. Each cell signal was acquired from the average of 256 spectra. The background signal from the BaF_2_ plates was obtained with 256 scans near the tissue sample. Spectra were acquired at a resolution of 2 ${cm}^{-1}$ in the range of 600–2500 ${cm}^{-1}$. Background signal from BaF_2_ was subtracted. Spectra were then further processed as described in Germond et al., 2018. Specifically, we performed a baseline correction using a polynomial fitting (5th order, 200 iterations) and vector normalisation. The area from 900 to 1780 ${cm}^{-1}$ was considered for subsequent machine learning analyses.

### 2.10. Statistical Analyses

Analysis of variance and comparisons of means were performed using one-way ANOVA tests followed by post hoc Tukey HSD tests at a significance threshold of *p* < 0.05. Statistical analyses were performed using Statistica software version 13.5 (Statistica).

#### Machine Learning Analyses

To visualise how spectral data are distributed in a low dimensional space, we used a t-SNE analysis [17]. Perplexity was adjusted to 50. A SVM model was computed to predict the 7 *postmortem* times from the sole spectral data (*n* = 805). The radial basis function (RBF) kernel was considered, and a 10-fold cross-validation was applied to train the model. The output of the model is a classification matrix showing the correct percentage of samples against the true value. Additionally, t-SNE and SVM models were performed in the Orange data mining software version 3.33 [18]. Then, to identify biomarkers of freshness or advanced *postmortem* time, we computed a linear discriminant analysis (DA) from PC components calculated on the same spectral data. The model assumed that each group had a normal distribution. The membership probabilities of each observation in a given group were determined using Mahalanobis distance (the distance of an observation from the mean of a group divided by the standard deviation along the direction vector). The F1 score was calculated from the sum of each principal component’s contribution, which is the product between the canonical scores and the weight associated with each PCA dimension. Here, the F1 score reflects the relative contribution of the wavelength in discriminating one or the other group, which helped us identify biomarkers attributed to freshness or D7 *postmortem* time. These analyses were performed in Jmp 11.0 (JMP, SAS Institute, Cary, NC, USA).

## 3. Results and Discussion

### 3.1. Postmortem Time Does Not Necessarily Impact Myofibrillar Protein Proteolysis and Solubility

Following slaughtering, *postmortem* processes cause profound changes over time in the structural and metabolic composition of animal tissues. This paper aimed to identify the possible biomarkers of *postmortem* aging time in trout muscles, along with *postmortem* kinetics. First, we investigated the evolution of the muscle’s biochemical composition over 15 days. At each time point, muscle samples were cut from six trout (Figure 1B) and subjected to multiple analyses (see Figure 2 for results). We observed that rigor mortis started around 12 h.

After the cessation of blood circulation and oxygen supply, *postmortem* metabolism leads to an accumulation of H^+^ in the muscle fibres, resulting in a drop in intracellular pH from approximately pH 7.0 to pH 6.0–6.5 in trout fillets, depending on the studies. Robb and colleagues [19] reported in trout that the pH decreased following death, starting from pH 6.7 ± 0.03 to below pH 6.6 after 2.5 h. They also showed the electro-stunning of fish caused a more rapid decrease than in anesthetised fish. Rammouz and colleagues [20] recorded a sharp decrease in pH from 6.6 to 6.17 over the first 24 h following the death in rainbow trout, but the values at 24 h ranged from 5.92 to 7.03.

Figure 2A shows that from Day 0 to Day 1, measured pH values of six trout were similar and not significantly different (6.4 ± 0.14 on D0 vs. 6.4 ± 0.08 on D1). The absence of a significant pH drop may reflect pre-slaughter stress, as the amplitude of the pH drop was reported to depend directly on the muscle glycogen content before slaughter [21,22]. In our study, the fish were loaded alive in the trailer and transported over 90 km in a truck, then unloaded at the processor plant before slaughter. This potentially caused unusual and stressful body movements, and therefore physical activity, which is expected to cause a strong glycogen depletion.

On Day 3 and later, pH values increased significantly up to 6.6 and then remained stable throughout *postmortem* time to Day 15 (ANOVA, Tukey HSD, *p* < 0.05; Figure 2A). However, we noted strong differences between fish (Figure 2A). This result is similar to previous studies, where the pH slowly increased after 24 h *postmortem* [19,20]. Hence, by comparison to these two studies, we observed a similar dynamic of pH variations. However, we found no particular explanation for the slight increase in pH in these papers following the rigor mortis. An increase in pH is typically associated with the metabolic activity of bacterial contamination [23], but here, the fish were kept on ice at 0 °C and did not show any sign of spoilage or unpleasant odor during the 15-day period. Hence, further work to explore the causes of pH increase would be valuable.

Lipid oxidation (Figure 2B) increased significantly (ANOVA, Tukey HSD, *p* < 0.05) up to D3, where the malondialdehyde value reached 3 mg/kg, then remained stable through Day 15. Therefore, this pattern of time course change overlaps with the time course change observed for pH. We did not note any rancid smell from the fish, suggesting no significant lipid oxidation. These results may be due to our good storage conditions on ice in a cold chamber, which keeps the fish close to 0 °C and prevents product degradation. Water activity (a_w_) decreased over time (Appendix A), which is expected with the loss of structure following product aging.

Concomitant with the rigor mortis, we were expecting the myofibrillar protein solubility and proteolytic activity to decrease (Figure 2D). However, values remained fairly stable to Day 15, and variations were not statistically significant (ANOVA, Tukey HSD, *p* < 0.05). This result is surprising. It is possible that the methods were not sensitive enough to detect significant changes. In meat, as in fish, *postmortem* proteolysis causes increased myofibrillar fragmentation and muscle tenderness [24]. In meat, studies showed that the myofibrillar protein index (MFI) increases significantly over time [25]. Our investigations using microspectroscopy (shown below) revealed significant changes attributed to proteolysis activity.

The colour of trout meat plays an important role in consumer purchase decisions. In this study, we used statistical analysis to verify whether *postmortem* time could affect the trichromatic parameters of raw fish (Figure 2E). Concomitant with the rigor mortis, lightness L*, and yellowness b*, increased over the first 24 h, contributing to a paler appearance of the fish. From Day 3 to Day 15, a* values remained stable and were not significantly different (one-way ANOVA, Tukey HSD, *p* < 0.05). A study conducted on salmon during 22 days of ice storage found only small differences in colour stability [26], which is in accordance with the present study.

Our results are consistent with previous studies conducted on fish [19], and on meat [27], showing that L* and b* increase following rigor mortis. Rammouz and colleagues [20] showed that high levels of activity or stress prior to slaughter can result in an increase in L* values. In addition, Robb and colleagues [19] reported that L* values were higher in electro-stunned fish than in anesthetised fish. The authors report that the reflection of light from the surface, measured by lightness, hue, and chroma, is caused by a loss in the flesh texture, rather than a loss in the pigment content (such as astaxanthin, which is a major pigment in rainbow trout). Rather, they link the observed trichromatic evolution to the denaturation of proteins, which become more insoluble and cause a loss of water from the flesh. We did observe a decrease in protein solubility from Day 0 to Day 1, although differences between days were not significant. We hypothesised that the absence of rapid pH drop during rigor mortis potentially limited the impact on protein solubility and denaturation. Consistent with this hypothesis, our data point to a good correlation between pH variations and L* values over a 15-day time course (*n* = 18 per time point, Sup. dataset), whereas Rammouz and colleagues [20] found no significant relationship between pH variations and luminance in trout.

Overall, the biochemical parameters measured showed little metabolic change in trout muscle during the 15-day *postmortem* time, at least in terms of protein proteolysis and lipid oxidation. This result is at least partially explained by the high activity of the trout before slaughter, and or our rigorous storage conditions at 0 °C.

### 3.2. Structural Postmortem Changes

To complement the above analyses and further explore the changes associated with *postmortem* time, we relied on complementary techniques, including structural observations and FTIR spectroscopy. Histological analyses and electron microscopy (Figure 3 and Appendix A) showed very good overall conservation of the muscle structures. Histological analyses of fibre structures in longitudinal muscle sections showed no significant changes over the first few days (Day 1 to Day 5) *postmortem* (Appendix A). This confirms the absence of protein denaturation and may suggest that the collagen content remains somewhat stable over the first few days. This result is partly supported by a previous study on trout stored at 5 °C, showing that the collagen content did not significantly change over the first three days [28]. The same study reported that the insoluble fraction of collagen increased afterward. In our study, we observed a loss of muscle–fibre attachment from Day 7 (Figure 3B), which was concomitant with a loss of the texture of the filet. A previous study by Taylor et al. [29] showed a loss of myofibre–myocommata attachment starting from 5 days of aging. Note that this structure modification is also known to occur within the first 24 h following freezing. Transversal breaks of the fibres were not observed at D9, but were observed at D15 (Figure 3C).

Ultrastructural analyses by transmission electron microscopy revealed that the sarcomeres, which were well-aligned at D0, started to lose their organisation at D7 (Figure 3D,E). The M lines, vertical black lines that are visible at D0, were misaligned but not yet disrupted at D7, and the intermyofibrillar spaces were increasing from D5. This result is in accordance with a previous study in sea bream, which showed that such alterations become visible from Day 5 and become progressively more severe thereafter up to Day 10 *postmortem* [30]. The sarcomere lengths average was 1608 nm (Day 0) and was significantly lower on Day 5, 7, and Day 15 (one-way ANOVA, Tukey HSD, *p* < 0.05) (Appendix A). We observed that starting D0, the sarcoplasm of myofibril cells included organelles such as mitochondria, which are characteristic of oxidative myofibres (Figure 3G–I).

### 3.3. Predicting Postmortem Day Using Spectral Data and Machine Learning

In addition to the above methods, we also explored the applicability of infrared spectroscopy combined with machine learning to predict the *postmortem* days from spectral data (Figure 4), and also to identify specific molecular bonds (i.e., biomarkers) associated with the freshness (Day 0) or with increased *postmortem* aging (Figure 5).

Figure 4A shows the averaged spectra for each time point. Typical spectral features of muscle cells were found, such as saturated lipids (1450 ${cm}^{-1}$), proteins (1650 ${cm}^{-1}$), carbohydrates, and nucleic acids (1740 ${cm}^{-1}$), along with smaller shoulder peaks associated with the secondary structures of proteins (ß-sheets, *α*-helix). To visualise the data in a low dimensional space, a t-SNE analysis was computed (Figure 4B). The analysis included five PC components. It shows the different days are separated, with the exception of Day 3 and Day 5, which tend to overlap. Interestingly, most of the time, the dispersion of the points in each group is smaller than between-day variations, which suggests the spectra hold valuable information to possibly predict the *postmortem* time. Thus, we computed an SVM model (10-fold cross-validation), using the seven time points as target classes (Figure 4C). The confusion matrix shown in Figure 4C shows that *postmortem* days were predicted very well, with the prediction accuracy ranging from 70.5% for Day 15 to 100% for Day 0 and Day 7. As far as we know, this is the first study to show the possibility of predicting *postmortem* time with such accuracy based on the sole information of label-free spectral measurements.

### 3.4. Spectral Biomarkers Associated with Muscle Freshness or Postmortem Aging

Industrial processors typically process fish up to 7 days after receiving them. To identify the metabolic shifts or biomarkers of freshness and *postmortem* aging, we used the spectra of cells measured at time points D0 and D7 to build a DA-PC classification model (Figure 4A) from which we can extract the canonical scores of each axis. One can then calculate the F1 score, which is interpreted as the contribution of a wavelength in discriminating one group or another. Previous studies (e.g., [31]) showed that this mathematical vector helps to find spectral biomarkers in an unsupervised and unbiased way from classification models, assuming the classification performs well. The F1 vector is shown in Figure 5C.

First, note that the nucleic acids-related molecular bonds, such as PO_2_ (1176 ${cm}^{-1}$, 1247 ${cm}^{-1}$), were strongly associated with fresh muscles (Figure 4D, Table 1). We expected to detect nucleic acids in fresh muscle. Nucleic acids are known to become degraded and not renewed in tissues following cellular death. Importantly, the intensity of the peak at 1176 ${cm}^{-1}$,was two-fold higher at D0 than at D7 (one-way ANOVA, *p* < 0.001). This makes it a relevant biomarker of freshness in fish muscles. Interestingly, Sánchez-Alonso et al. [32] previously used Raman spectroscopy, a vibrational spectroscopy technique complementary to FTIR, to monitor the effect of frozen storage on hake fillets. They showed that freezer storage caused a disappearance of the PO_2_^−^ band, which they attributed to membrane lipid hydrolysis and toughening of the fish flesh.

We also identified wavelengths associated with amide I α-helix and ß-sheet structures (Table 1). These important structures determine protein conformation and integrity. According to the F1 vector, the peak at 1553 ${cm}^{-1}$, which is associated with α-helix structures, strongly contributed to identifying fish freshness. ANOVA tests confirmed that the peak intensities at 1553 ${cm}^{-1}$,and 1635 ${cm}^{-1}$ were significantly higher at D0 than D7 (one-way ANOVA, *p* < 0.002), suggesting that either protein content increased or that the proteins did not undergo denaturation. Moreover, the intensity of the 1760 ${cm}^{-1}$ band significantly decreased with *postmortem* aging (Figure 4A), while strong intensity values contributed to identifying the D0 sample. This band contribution was confirmed in the F1 vector (Figure 4D, Table 1). We associated this band with the presence of glycogen and lipids (Table 1).

Antiparallel ß-sheets are usually thought of as being a stable structure originating from native protein. Does the *postmortem* aging lead to an aggregation of proteins or antiparallel ß-sheets, leading to a higher relative contribution of antiparallel ß-sheets in the spectral signature of D7 tissues? Further investigations are needed to verify this phenomenon, and there may be other confounding factors that explain the contribution of these two bands, such as storage conditions. In the literature, these two markers (1490 and 1670 ${cm}^{-1}$), were reported to be impacted by storage conditions, such as freezing [10].

In addition to the above work, we also performed the same analysis but by using Day 0 versus Day 15 (Appendix A, Appendix A). The biomarkers found confirmed the contribution of nucleic acid-related bands (1176 ${cm}^{-1}$, 1247 ${cm}^{-1}$) and other bands in discriminating the fresh muscle tissues. Peaks that helped identify D7, such as 1490 and 1670 ${cm}^{-1}$ (Table 1), also contributed to identifying the aging state at D15 (Appendix A, Appendix A). We thus conclude that the disappearance of nucleic acids and an increase in fatty acids in the spectral signatures of *postmortem* tissues may potentially be useful predictors of the advancement of *postmortem* aging.

We explored how well biomarkers of *postmortem* aging could correlate with the biochemical parameters shown in Figure 2. For this analysis, we randomly divided our spectral dataset into three so as to match the number of physicochemical measurements, thus enabling us to calculate correlations (Appendix A). Interestingly the protein solubility values shown in Figure 2C correlated negatively at a significant level (*p* ≤ 0.01) for nucleic acids (1247 ${cm}^{-1}$) and CH_2_ bending in saturated lipids (1469 ${cm}^{-1}$), which are markers of freshness (D0). A positive and significant correlation between the ß-sheet secondary structures of amide I (1670 ${cm}^{-1}$) and solubility was found (*p* < 0.05). This shows that FTIR is able to monitor protein solubility with good confidence at the fibre level. On the other hand, proteolysis values did not correlate with major spectral peaks in a significant manner. There are few correlations between spectral data and pH or colour parameters (Appendix A).

In addition, we explored how the ratios between the major spectral peaks identified in Table 1 differed between D0 and D7. Our rationale was that ratiometric analyses can provide reliable indicators of freshness that should remain consistent across different FTIR measurement platforms. These analyses are shown in Appendix A. These results highlight in particular that normalising major bands against the 1186 ${cm}^{-1}$ peak (associated with amide III) enables the distinguishing of the fresh fish from fish at D7 *postmortem* time. Future work should evaluate the variability of such ratios across a large population of calibrated fish.

Finally, to state one limitation of this study, we did not study how *postmortem* biomarkers could vary along the head-to-tail axis or back-to-ventral axis. In a recent previous study by our team, we showed that lipid distribution in salmon decreases from head to tail [33], in accordance with scientific literature [34,35,36,37]. In salmon, lipid content decreases in the white muscle when going from the dorsal fat depot to the backbone, but then increases in the belly flap according to previous results [37,38,39]. However, as far as we know, protein denaturation is not different along the head-to-tail axis over *postmortem* maturation.

## 4. Conclusions and Perspectives

This study highlighted biochemical and structural changes during a span of 15 days of *postmortem* aging in trout muscle using complementary approaches. A combination of machine learning algorithms with label-free spectral characterisation calibrated and trained to selected wavelengths could accurately predict *postmortem* aging. This work raises important prospects to solve major societal issues regarding filet processing. In particular, in light of our results, it will be important to investigate how *postmortem* aging will impact the diffusion of salt and its final concentration in products.

## Figures and Tables

**Figure 1 foods-12-01957-f001:**
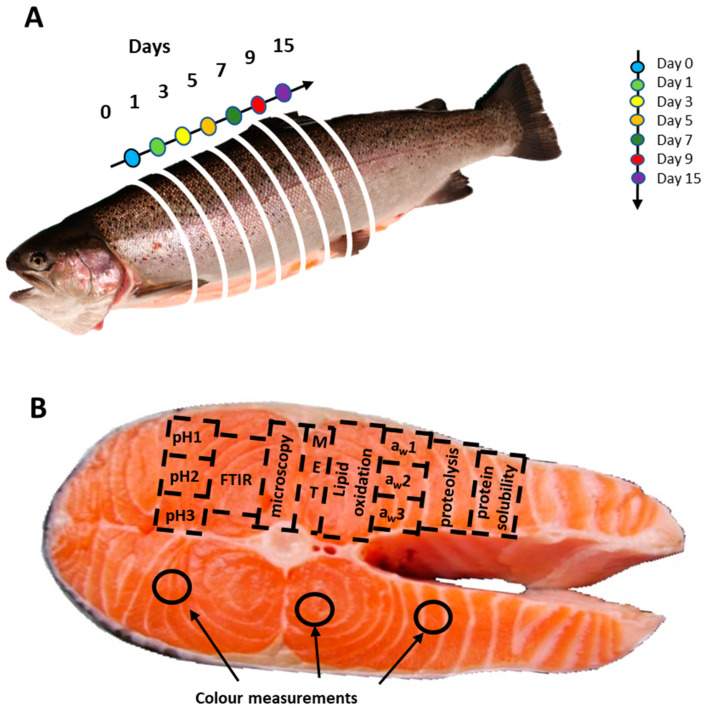
Overview of our measurement locations for sampling on trout. (**A**) Measurements were performed at various *postmortem* times over fifteen days (D0 to D15). (**B**) Sampling for a given technique was conducted at the same location of a slice, and slices are different between days to enable the longitudinal study.

**Figure 2 foods-12-01957-f002:**
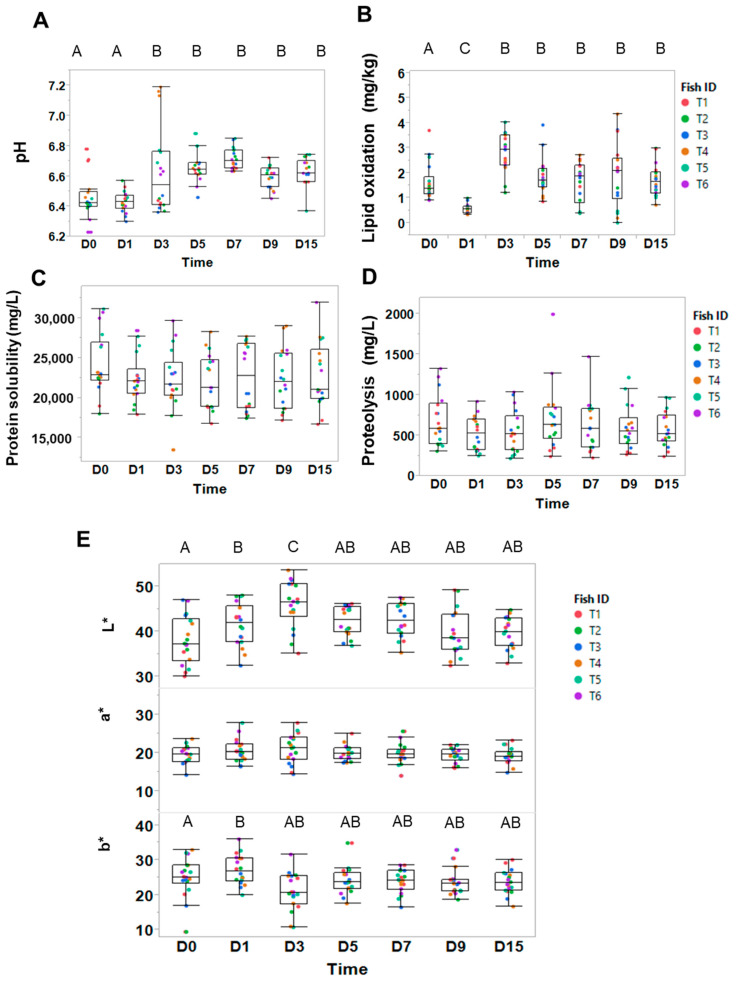
Physicochemical parameters of trout muscles measured at various *postmortem* time points over 15 days (D0 to D15). Measurements of (**A**) pH, (**B**) lipid oxidation, (**C**) protein solubility, (**D**) proteolysis, and (**E**) colour parameters (L* stands for Luminance, a* stands for redness and b* yellowness). Measurements were performed in triplicate on six filets (*n* = 18 per time point). A colour bar was attributed to each fish to visualise variabilities across individuals. ANOVA followed by posthoc HSD Tukey tests were performed. When present, capital letters show statistical differences that are significant between groups (*p* < 0.05).

**Figure 3 foods-12-01957-f003:**
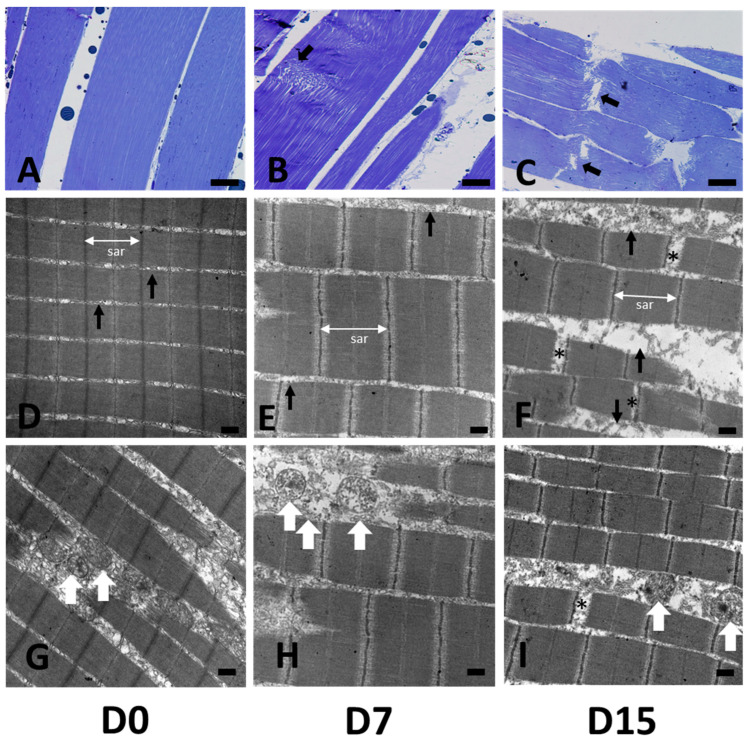
*Postmortem* changes in trout muscle micro and ultrastructures. From left to right: Histological analysis of muscle fibres at (**A**) D0, (**B**) D7, and (**C**) D15. Bar scale: 50 μm. White arrows show loss of myofibrillar arrangement, which first appeared at D7. Black arrows show fibre ruptures appearing from D15. (**D**–**F**) (from left to right): electron microscopy images of muscle ultrastructures at D0, D7, and D15, respectively. Bar scale: 0.5 μm. The white bar labelled “sar” shows the length of sarcomere fibres. Black arrows (**D**–**F**) show the intermyofibrillar space, visible after D7. Asterisks show myofibrillar ruptures. White arrows (**G**–**I**) reveal the presence of mitochondria between myofibrils.

**Figure 4 foods-12-01957-f004:**
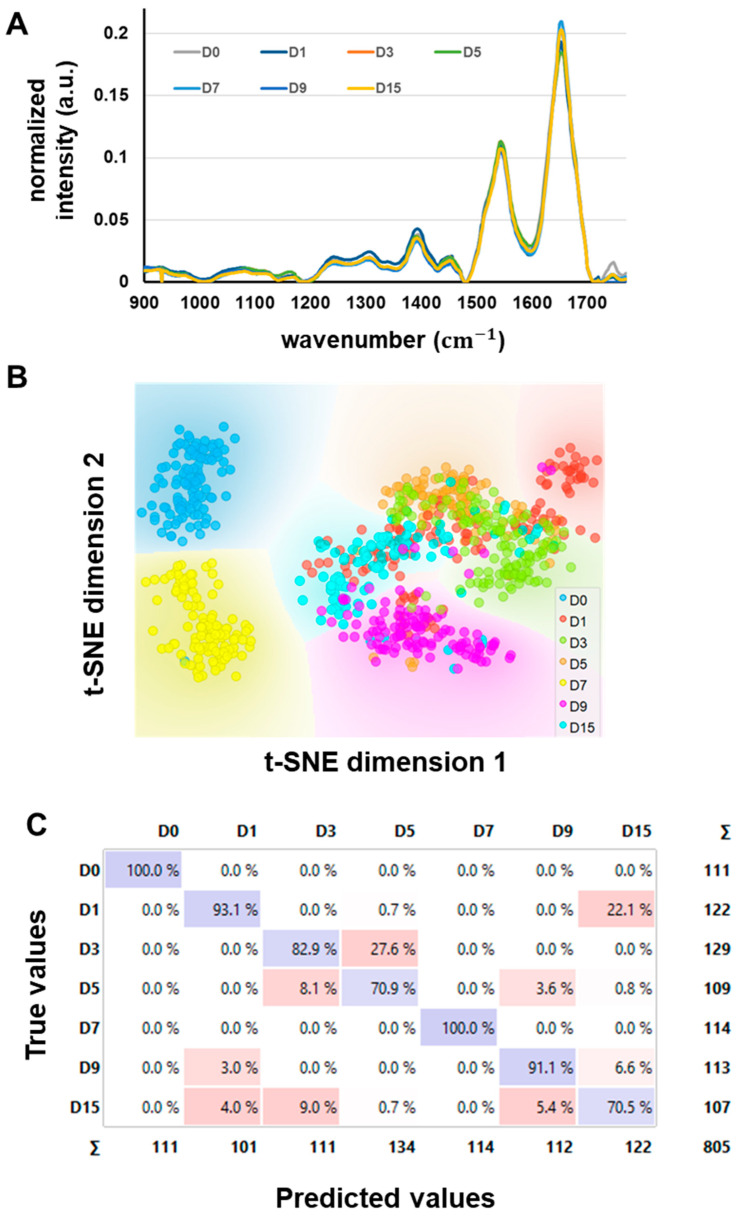
Infrared microspectroscopy analysis of single cells in *postmortem* muscle tissues and prediction of *postmortem* time. (**A**) Average spectra of the fingerprint region (900–1780 ${cm}^{-1}$) of single fibres at the different *postmortem* time points (total *n* = 805 spectra). (**B**) t-SNE visualisation of the differences between *postmortem* days. (**C**) Confusion matrix of the SVM model, showing the strong ability of SVM in predicting correctly the *postmortem* days.

**Figure 5 foods-12-01957-f005:**
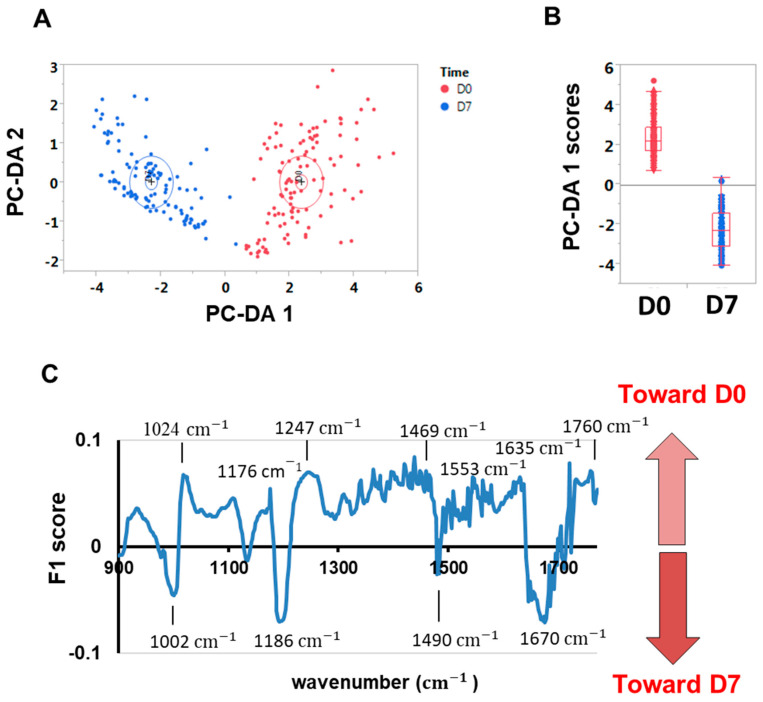
Identification of biomarkers for freshness and Day 7 *postmortem* time. (**A**) To visualise the difference between Day 0 and Day 7, a PC-DA analysis was performed in which each point corresponds to a single fibre spectrum (*n* = 225). For D0, 111 single fibres were measured. For D7, 114 single fibres were measured. (**B**) Canonical scores along the first axis show Day 0 and Day 7 spectral information is easily distinguishable. (**C**) F1 score reveals positive values that are mostly associated with Day 0, and negative values that are mostly associated with Day 7. Highlighted wavenumber values correspond to well-known biological compounds (see Table 1).

**Table 1 foods-12-01957-t001:** Molecular bonds were identified from the classification model of fresh muscles (D0) over more advanced *postmortem* time (D7). Biological compounds associated with these spectral ranges were identified from the literature. A number of the biomarkers found for D7 were also identified at D15 in a separate model comparing D0 to D15 (Appendix A).

Wavenumber	Biomarker For	Molecular Bond	Biological Compounds
1002 ${cm}^{-1}$	D7	Phe, Tyr	aromatic compound, protein
1024 ${cm}^{-1}$	D0	CO	polysaccharides, glycogen
1172–1176 ${cm}^{-1}$	D0	CC of DNA, PO_2_ stretch	DNA, PO_2_, nucleic acids, carbohydrates
1186 ${cm}^{-1}$	D7	amide III	amide III
1204–1206 ${cm}^{-1}$	D7	amide III	amide III, collagen
1247 ${cm}^{-1}$	D0	PO_2_^−^	nucleic acids (RNA)
1363 ${cm}^{-1}$	D0	CH_2_, CC	polysaccharides
1469 ${cm}^{-1}$	D0	CH_2_ bending of lipids	saturated lipids
1490 ${cm}^{-1}$	D7	CC, CH	amide I, fatty acids
1553 ${cm}^{-1}$	D0	amide II	predominantly α-helix of amide II
1635 ${cm}^{-1}$	D0	ß-sheet structures of amide I	ß-sheets, proportion of secondary protein structures (shoulder peak of amide I)
1670 ${cm}^{-1}$	D7	𝒱(CC) *trans*, antiparallel ß-sheet structures of amide I	antiparallel ß-sheets (shoulder peak of amide I), fatty acids
1712 ${cm}^{-1}$	D0	CO	glycogen, lipid oxidation, unsaturated fatty acids (PUFA)
1760 ${cm}^{-1}$	D0	CO, COO-R	glycogen, lipids

Bands at 1490 and 1670 ${cm}^{-1}$, associated with carbon double bonds, antiparallel ß-sheets, and fatty acids, were significantly higher at D7 than D0 (one-way ANOVA, *p* < 0.001).

## Data Availability

The datasets generated for this study are available on request to the corresponding author.

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
