# Peer review of "The Effects of Postmortem Time on Muscle Trout Biochemical Composition and Structure"

_foods, 2023, doi:10.3390/foods12101957_

Round 1

Reviewer 1 Report

This manuscript study several postmortem biochemical and structural characteristics of trout muscles during the 15-day period following animal slaughtering. The content is rich and interesting. It has important to solve major societal issues regarding filet’s processing. But I have the following concerns about the manuscript

1. Line 4, Is Pierre-Sylvain Mirade and Thierry Astruc the same author? If not, please add “a” after Pierre-Sylvain Mirade and change “and” to “,”.

2. Line 6, Correspondence shouldn't be superscript. And “Correspondence:” should be revised by “Correspondence”.

3. Lines 114 and 126, There should be Spaces between numbers and units (3 M, 1 g).

4. Line 246, It can be seen from Figure 1A that the measurement locations for sampling of different days are different, which will lead to discredited result?

5. Line 251, Personally, I don't think the title should be written like this. It can be changed to “the effect of postmortem time on myofibrillar protein proteolysis and solubility”, and stuff like that.

6. Lines 387, 364-366, Why is Figure 3 D15 better than Figure 3D0 and D7? This seems to contradict that “These organelles showed progressive degradation and structural disorganization as exemplified on days 7 and 15 (Figure 3F, H, I).”

Examine the manuscript carefully to avoid any fundamental errors. Like: no Spaces between units and numbers, whether superscript and so on.

Author Response

We thank reviewer 1 for his time and most valuable comments. Importantly, Figure 3 was improved. Errors were found in the text and corrected. English was also improved. A new title is suggested, as to comply with the reviewer’s request. We hope these modifications are satisfactory.

Reviewer #1

This manuscript study several postmortem biochemical and structural characteristics of trout muscles during the 15-day period following animal slaughtering. The content is rich and interesting. It has important to solve major societal issues regarding filet’s processing. But I have the following concerns about the manuscript:

  1. Line 4, Is Pierre-Sylvain Mirade and Thierry Astruc the same author? If not, please add “a” after Pierre-Sylvain Mirade and change “and” to “,”.
  2. Line 6, Correspondence shouldn't be superscript. And “Correspondence:” should be revised by “Correspondence”.

These points have been addressed in the revised manuscript & in the Supplementary data file. We changed Correspondence to “Corresponding authors”, is this more suitable?

  1. Lines 114 and 126, There should be Spaces between numbers and units (3 M, 1 g).

Thanks for noticing, these points have been addressed in the revised manuscript and highlighted 3 corrections in red.

  1. Line 246, It can be seen from Figure 1A that the measurement locations for sampling of different days are different, which will lead to discredited result?

Thank you for your comment. To prevent any variability, sampling for a given analytical technique was done on the same location of a slice, and slices are indeed different between days to enable a longitudinal study, as mentioned in M&M. This was added in Figure 1’s legend. This does not lead to discredit results, but a few limitations which were already addressed line 522. However, this sampling strategy enables to compare the same portion areas of muscles from one day to another.

  1. Line 251, Personally, I don't think the title should be written like this. It can be changed to “the effect of postmortem time on myofibrillar protein proteolysis and solubility”, and stuff like that.

Thanks for this suggestion. We suggest the following title to « the effects of postmortem time on muscle trout biochemical composition and structure.”

  1. Lines 387, 364-366, Why is Figure 3 D15 better than Figure 3D0 and D7? This seems to contradict that “These organelles showed progressive degradation and structural disorganization as exemplified on days 7 and 15 (Figure 3F, H, I).”

We agree that the pictures chosen as Fig 3 D0 D7 and D15 were not representatives of the general structures’ arrangements. Hence, we propose new pictures for G H and I. In these, structures are still aligned at D0 (G), and start to be misaligned at D7 (H) and D15 (I). This is a better representative of Figures 3D, E, and F. In Fig 3D GHI, the mitochondria are visible within structures which we wanted to highlight in this part of the results. To avoid any confusion, we removed the sentence “These organelles showed progressive degradation and structural disorganization as exemplified on days 7 and 15” which was written to denote the general structure.

Reviewer 2 Report

The experiment involved recognizing changes in trout meat during storage using spectral imaging. The Authors have also attempted to adapt this method to estimate the length of post-mortem time when the buyer has no information about it. The research performed was well justified. The experiment was very well prepared, and the analysis methods were described in sufficient detail. The results were well presented. I rate the work highly, but I have a few comments, which I will discuss below.

1)      In the “Introduction”, in lines 47-52, the Authors provided information that biochemical analyses conventionally performed in the industry are affected by various factors, and processing industries face a large diversity of products. Did the authors mean that the method they are developing (FTIR), compared to those used conventionally, is universal and free from such a defect? Probably not, and in that case, I would suggest removing this part of the text. If I'm wrong, please expand on this thought regarding using the FTIR method.

2)      In the “Introduction,” I would not only mention the influence of ripening time on the quality characteristics of fish meat and thus consumer decisions but also the salting process, which is discussed in the “Conclusion and Perspective” subchapter.

3)      Lines 257 – 258: „rigor mortis started after twelfth (or 12.) hour” or “…after 12 hours” instead of “rigor mortis started around 12 hours”. “Rigor mortis" should be in italics.

4)      Line 277: “fish” instead of “fishes”

5)      Line 411: Probably “shown in Figure 5C” should be instead of “shown in Figure 4D”. The same is in lines 413 and 431.

6)      In lines 431-432, there is information that the Authors associate the 1760 cm-1 band with the presence of glycogen and lipids, but in Table 1, only glycogen is listed in the “Biological compounds” column.

7)      Lines 460 – 462: Authors state that peak 1490 helped identify the D7 and D15 aging state and refer to Table 1 in Supplementary data, but there is no such wavenumber in Supplementary Table 1.

8)      Lines 469-473: Authors stated that protein solubility values correlated for three peaks, 1186 cm-1, 1670 cm-1, and 1469 cm-1, and referred to Sup. Figure 4, but this figure shows correlations only for 1469 cm-1 and 1247 cm-1(not mentioned in the text). 1186 cm-1 and 1670 cm-1 are missing.

9)      Line 472: “Sup. Figure 4 in Supplementary dataset” instead of “Sup. Figure 4 and Supplementary dataset.”?

10)   Line 473: “tested” instead of “testes”

11)   Line 482: “fish” instead of “fishes”

12)   Line 479: “filet processing instead of “filet’s processing”

13)   After moving the information about salting fish to the “Introduction,” you can leave a one-sentence mention about it in the “Conclusion and Perspectives” subchapter.

14)   Supplementary Table 1: Strangely, the band at 1363 cm-1 is assigned as a biomarker for D15. The same peak is identified in Table 1 as a biomarker for D0. Also, the 1670 cm-1 marker in Supplementary Table 1, I think, should be for D15 instead of D7.

There were just a few minor errors in the manuscript (see the comments in the review). It is worth reviewing the text again regarding linguistic correctness to eliminate imperfections.

Author Response

We thank reviewer 2 for his time and most valuable comments and kind corrections. Importantly, Figure 3 was improved. The sup Excel file was added. Errors were found in the text and corrected. English was also improved. A new title was suggested, as to comply with reviewer 1’s request. We hope these modifications are satisfactory.

#Reviewer 2

There were just a few minor errors in the manuscript (see the comments in the review). It is worth reviewing the text again regarding linguistic correctness to eliminate imperfections.

The experiment involved recognizing changes in trout meat during storage using spectral imaging. The Authors have also attempted to adapt this method to estimate the length of post-mortem time when the buyer has no information about it. The research performed was well justified. The experiment was very well prepared, and the analysis methods were described in sufficient detail. The results were well presented. I rate the work highly, but I have a few comments, which I will discuss below.

1)      In the “Introduction”, in lines 47-52, the Authors provided information that biochemical analyses conventionally performed in the industry are affected by various factors, and processing industries face a large diversity of products. Did the authors mean that the method they are developing (FTIR), compared to those used conventionally, is universal and free from such a defect? Probably not, and in that case, I would suggest removing this part of the text. If I'm wrong, please expand on this thought regarding using the FTIR method.

This critic is right and thus we removed this part of the text. Initially, we wanted to highlight that many factors (species, age, seasons)… impact product diversity, which is a challenge in the processing industries. The promise of spectral imaging is to provide a label-free and rapid way for characterization, rather than overcoming hurdles seen with other techniques. To clarify this point, modifications were done and highlighted in red in the text.

2)      In the “Introduction,” I would not only mention the influence of ripening time on the quality characteristics of fish meat and thus consumer decisions but also the salting process, which is discussed in the “Conclusion and Perspective” subchapter.

Thank you, this point was added.

3)      Lines 257 – 258: „rigor mortis started after twelfth (or 12.) hour” or “…after 12 hours” instead of “rigor mortis started around 12 hours”. “Rigor mortis" should be in italics.

Thank you, this point was corrected.

4)      Line 277: “fish” instead of “fishes”

Thank you, it was corrected.

5)      Line 411: Probably “shown in Figure 5C” should be instead of “shown in Figure 4D”. The same is in lines 413 and 431.

Thank you for noticing, the label of Figure 4D was wrong, and replaced by Figure 5C.

6)      In lines 431-432, there is information that the Authors associate the 1760 cm-1 band with the presence of glycogen and lipids, but in Table 1, only glycogen is listed in the “Biological compounds” column.

Thank you for noticing, the table was modified and molecular assignments were verified.

7)      Lines 460 – 462: Authors state that peak 1490 helped identify the D7 and D15 aging state and refer to Table 1 in Supplementary data, but there is no such wavenumber in Supplementary Table 1.

Thank you for noticing, the table in Sup. Data was modified after verification.

8)      Lines 469-473: Authors stated that protein solubility values correlated for three peaks, 1186 cm-1, 1670 cm-1, and 1469 cm-1, and referred to Sup. Figure 4, but this figure shows correlations only for 1469 cm-1 and 1247 cm-1(not mentioned in the text). 1186 cm-1 and 1670 cm-1 are missing.

Thank you, the text should refer not to Figure 4 but to a supplementary Excel file where we report all correlation values. (Supplementary Excel file) which file is uploaded with the revision. We also noticed errors in the text due to confusion with Sup.Fig.4. The following modification for done:

“Interestingly the protein solubility values shown in Figure 2C correlated negatively, at a significant level (p ≤ 0.01) for nucleic acids (1247 cm-1), and CH2 bending in saturated lipids (1469 cm-1), which are markers of freshness (D0). A positive and significant correlation between the ß-sheet secondary structures of amide I (1670 cm-1) and solubility was found (p ≤ 0.05). This shows FTIR is able to monitor protein solubility with good confidence at the fiber level. On the other hand, proteolysis values did not correlate with major spectral peaks in a significant manner. Few correlations between spectral data and pH or colour parameters (Supplementary Excel file).”

9)      Line 472: “Sup. Figure 4 in Supplementary dataset” instead of “Sup. Figure 4 and Supplementary dataset.”?

Thank you this was corrected.

10)   Line 473: “tested” instead of “testes”

This sentence was removed with the above modifications.

11)   Line 482: “fish” instead of “fishes”

Thank you this was corrected.

12)   Line 479: “filet processing instead of “filet’s processing”

Thank you this was corrected.

13)   After moving the information about salting fish to the “Introduction,” you can leave a one-sentence mention about it in the “Conclusion and Perspectives” subchapter.

The information about salting was moved in the introduction as recommended, and the following sentence was left in the « Conclusion & Perspective »: In particular, in light of our results, it will be important to investigate how postmortem aging will impact the diffusion of salt and its final concentration in products.

14)   Supplementary Table 1: Strangely, the band at 1363 cm-1 is assigned as a biomarker for D15. The same peak is identified in Table 1 as a biomarker for D0.

Indeed, this peak contribution is different in the 2 models’ F1 scores, clearly positive for F1 in Figure 5 and negative for Sup. Figure 3. This difference arises due to the fact D0 vs D7 and D0 vs D15 are different low-dimensional spaces (as mentioned in Table 1’s legend). D7 and D15 are very different (Figure 4C, separated at 100%) explaining different relative contributions for a few particular peaks. The result remains valid from a statistical / modelling view point. To keep a clear line of thought, in the manuscript (line 492-499) we have focused on the major peaks that shows similar tendencies between D7 and D15. However, how peaks’ contribution varies in these PCA-DA low dimensional models is an open question. From a personal viewpoint, I envision the ratiometric analyses might identify more consistent trend/biomarkers across postmortem days, fish, and platforms but this remains to be tested.

Also, the 1670 cm-1 marker in Supplementary Table 1, I think, should be for D15 instead of D7.

Thank you for noticing, the label was corrected.
